



# Temperature-dependent rate coefficients for the reaction of OH radicals with selected alkanes, aromatic compounds and monoterpenes

Florian Berg[1], Anna Novelli[1], René Dubus[1], Andreas Hofzumahaus[1], Frank Holland[1], Andreas Wahner[1], and Hendrik Fuchs[1,2]

[1]Institute of Climate and Energy Systems, ICE-3: Troposphere, Forschungszentrum Jülich GmbH, Jülich, Germany
[2]Department of Physics, University of Cologne, Cologne, Germany

**Correspondence:** Hendrik Fuchs (h.fuchs@fz-juelich.de) and Anna Novelli (a.novelli@fz-juelich.de)

**Abstract.** The rate coefficients of the reaction of hydroxyl radicals (OH) with 12 different volatile organic compounds (VOCs), methane, ethane, propane, n-butane, methyl vinyl ketone (MVK), $\gamma$-terpinene, $\Delta^3$-carene, myrcene, toluene, o-xylene, m-xylene, mesitylene were studied in an absolute rate study in $1\,\text{atm}$ of air between 280 and $340\,\text{K}$ using an OH reactivity instrument with a temperature controlled flow tube. There are few or no measurements in the literature for the important

monoterpene and aromatic compounds emitted by biogenic and anthropogenic sources, although the oxidation of these species is particularly important in the formation of secondary pollutants such as ozone and particles. The time-dependent decay of the OH concentration was measured in a flow tube, allowing the pseudo-first order rate coefficient to be determined after the production of a moderately high OH concentration (about $1\cdot10^9\,\text{cm}^{-3}$) by photolysis of ozone in the presence of water vapour. In contrast to many previous studies, the OH reaction took place in a volume at ambient conditions, while the detection of OH

was achieved by fluorescence in a separate low pressure detection volume. The VOC concentrations were measured using the total organic carbon method. With the methods used, a high accuracy of the rate constants is obtained (2-$\sigma$ uncertainties approximately $6\,\%$). The uncertainties of the values are in most cases smaller than those of values recommended by IUPAC and NASA-JPL or reported in the literature, where available.

## 1 Introduction

The hydroxyl radical (OH) is the most important oxidant in the atmosphere, transforming inorganic and organic in the gas phase (Lelieveld et al., 2004). Volatile organic compounds (VOCs) are a diverse group of carbon-containing compounds that are mainly in the gas-phase for tropospheric conditions. They have natural and anthropogenic sources, with natural emissions coming mainly from vegetation, while anthropogenic sources include industrial processes, vehicle emissions, and solvent use. VOCs have a major impact on air quality, as their oxidation leads to the formation of ground-level ozone (Haagen-Smit, 1952)

and secondary organic aerosols (Kroll and Seinfeld, 2008), both of which also affect the Earth's climate.

Isoprene, a $C_5$ conjugated diene ($C_5H_8$), dominates the total emissions of biogenic VOCs with monoterpenes ($C_{10}H_{16}$) and sesquiterpenes ($C_{15}H_{24}$) also playing an important role (Guenther et al., 2012). These classes of unsaturated hydrocarbons



include thousands of species, but approximately 20 species are often found in significant amounts in the atmosphere, e.g.
$\alpha$-pinene, $\Delta^3$-carene, $\beta$-pinene, $\delta$-limonene, myrcene, camphene, ocimene, sabinene (Guenther, 2015). Anthropogenic non-
methane VOCs from fossil fuel combustion and volatile chemical product (VCP) evaporation are most significant in urban and
industrial areas (Piccot et al., 1992; Coggon et al., 2021; Gkatzelis et al., 2021). The monocyclic aromatic compounds benzene,
toluene and xylene isomers and trimethylbenzenes such as mesitylene are some of the important species from fossil fuel and
solvent use (Ponnusamy et al., 2017; Masih et al., 2018).

A large number of kinetic studies of the reaction of VOCs with OH have been carried out in the past. However, there are
still gaps in the knowledge, especially regarding the temperature dependence of reaction rate constants, which is particularly
important for implementation in chemical transport models. To date, there are only few studies on the temperature dependence
of the rate coefficients for the reaction of OH radicals with monoterpenes such as $\Delta^3$-carene, myrcene and $\gamma$-terpinene and
methyl vinyl ketone (MVK, Kleindienst et al. (1982); Gierczak et al. (1997a)), one of the major oxidation products of isoprene.
Furthermore, only a few studies have been carried out for aromatic species from anthropogenic sources (toluene, Semadeni
et al. (1995); Tully et al. (1981), mesitylene, Aschmann et al. (2006); Bohn and Zetzsch (2012); Alarcon et al. (2015), m-xylene
and o-xylene, Nicovich et al. (1981)).

A wide range of laboratory methods including absolute and relative rate studies have been used in the past to study the kinet-
ics of OH reactions in the gas phase (e.g. Atkinson, 1986; Kurylo and Orkin, 2003). One of the most powerful techniques uses
flash photolysis to produce OH radicals from a photolytic precursor, combined with fluorescence detection of the OH decaying
in the presence of an excess of reactant molecules. Under pseudo-first order conditions, the OH decay can be described by a
single exponential function and the absolute OH reaction rate constant can be determined from the decay rate and the reactant
concentration.

Based on this method, rate constants for the reaction of OH with aromatics and olefinic VOCs have been determined using
flash lamp photolysis resonance fluorescence (FP-RF, e.g., Hansen et al. (1975); Perry et al. (1977); Ravishankara et al. (1978);
Nicovich et al. (1981); Tully et al. (1981); Kleindienst et al. (1982); Bohn and Zetzsch (2012); Alarcon et al. (2015)) or laser
photolysis laser-induced fluorescence (LP-LIF, e.g., Gierczak et al. (1997a); Dillon et al. (2017)). In the previous studies, the
photolytic OH production and fluorescence detection took place in the same volume. The total pressure in the volume was kept
low to minimise quenching of the OH fluorescence by molecular collisions and thus allow sensitive OH detection.

In this study, a method was applied which has previously been used for OH reactivity measurements in the field (Sadanaga
et al., 2004; Lou et al., 2010; Yang et al., 2016; Fuchs et al., 2017). Here, OH reactivity means the inverse atmospheric lifetime
of OH in air containing reactive pollutants. OH reactivity instruments have also been used in kinetic experiments, mainly to
test their instrumental performance by measuring well-established rate constants of OH reactions, e.g. with carbon monoxide
(CO), methane ($CH_4$) and propane ($C_3H_8$) (Sadanaga et al., 2004; Amedro et al., 2012; Nakashima et al., 2012; Stone et al.,
2016; Pang et al., 2023). For the measurements in this work, an OH reactivity instrument was further developed to investigate
the temperature dependence of OH reaction rate constants.

In order to measure a specific reaction rate constant, humidified synthetic air containing some ozone ($O_3$) and a known
concentration of a single reactant is sampled into a temperature-controlled flow tube at atmospheric pressure. In the tube, OH




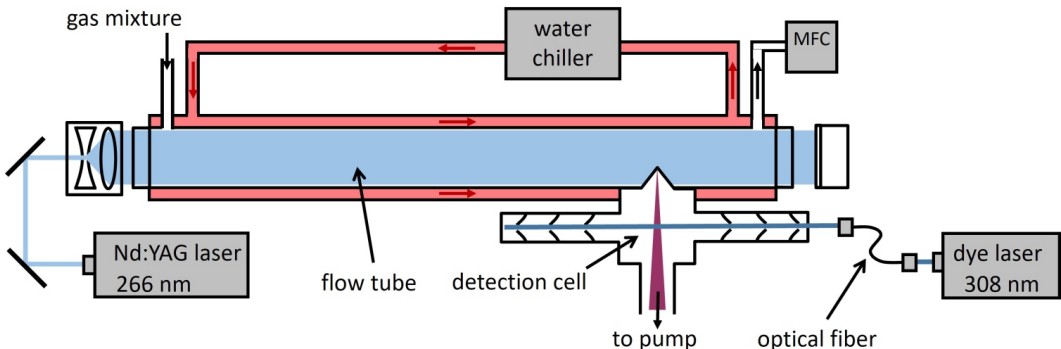

**Figure 1.** Schematic of the OH reactivity instrument used to determine the temperature dependent rate coefficients of the OH reaction with volatile organic compounds. A well-defined mixture of humidified air with the reactant and a low concentration of ozone is flowed through a flow tube, in which OH radicals are produced by laser flash photolysis of ozone at $266\,\mathrm{nm}$. The decay of the OH concentration is observed by laser-induced fluorescence using $308\,\mathrm{nm}$ radiation from a dye laser.

is produced by laser flash photolysis (LP) of ozone at $282\,\mathrm{nm}$ and the subsequent OH decay is monitored in real time by laser-induced fluorescence (LIF) using laser pulses at $308\,\mathrm{nm}$ (Lou et al., 2010). Unlike previous laboratory set-ups, where the

60 chemical reaction of OH and its detection took place in the same volume, in the current instrument OH detection is achieved in a separate low-pressure detection cell which continuously draws air from the reaction tube. The method of OH detection by gas-expansion and low-pressure LIF is highly sensitive (Hofzumahaus et al., 1996). It allows kinetic studies at low radical concentrations of $10^6$ to $10^9\,\mathrm{cm^{-3}}$) at atmospheric conditions, avoiding interference from secondary chemistry.

  In this study, the temperature-dependent OH rate constants were determined for the reactions of n-alkanes (methane, ethane,

propane, n-butane) to test and validate the method and of eight other VOCs (methyl vinyl ketone, $\Delta^3$-carene, $\gamma$-terpinene, myrcene, toluene, mesitylene, m-xylene, o-xylene), which have large biogenic or anthropogenic emission sources and for which only few or no reaction kinetic studies are available.

## 2 Methods

### 2.1 Measurement of the pseudo-first order rate coefficient of OH

The experiments were carried out using a modified laser flash-photolysis laser-induced fluorescence (LP-LIF) instrument for OH reactivity measurements. It consists of a flow tube and an OH measurement section (Fig. **??**).

  In the experiments, small flows of ozone in synthetic air (flow rate $100\,\mathrm{cm^3min^{-1}}$) and of a OH reactant in synthetic air (flow rates in the range of a few $10\,\mathrm{cm^3min^{-1}}$) were added to humidified synthetic air mixed from evaporated liquid nitrogen and oxygen (purity $> 99.9999\,\%$, Linde) and sampled into the flow tube. The total flow rate was $20\,\mathrm{Lmin^{-1}}$), resulting in a



residence time of less than 2 s. All flows were controlled by calibrated mass flow controllers, which gave an accuracy of the flow rates to within 0.75 % of the reading.

In previous versions of the OH reactivity instrument, temperature and pressure of the flow tube could not be systematically changed (Lou et al., 2010). Here, the flow tube is made of a double walled stainless steel tube coated with SilcoNert ® to prevent loss of OH reactants on the wall. Its temperature could be controlled by flowing temperature-controlled water around 80 the flow tube. This allowed the air temperature to be varied within a range of 280 to 340 K, which could be extended in the future, if a different liquid is used. In addition, before entering the flow tube, the air was heated or cooled by firstly passing the pure synthetic air through a Teflon tube inserted into the water chiller and secondly by passing the air mixture through a large diameter section of the double walled flow tube, where the photolysis laser is not applied. This ensured that the air was at the same temperature as the flow tube before entering the photolysis zone.

The air temperature inside the flow tube was monitored by 2 calibrated PT-100 temperature sensors located at the inlet and outlet of the flow tube. Both sensors gave the same readings, demonstrating that the air temperature did not change along the length of the flow tube. Thermal equilibrium of the entire system was reached after approximately 20 min, when the temperature was reduced by 10 K. The time needed to reach thermal equilibrium was longer when the temperature difference was higher or the temperature was increased. Therefore, the temperature dependence of the OH rate coefficients was measured 90 by decreasing the temperature in steps of 10 K starting from the highest value. During the time it took to reach thermal equilibrium, the zero-decay rate coefficient was measured by switching the small flow containing the OH reactant to a vent line using a Teflon solenoid valve.

The water vapour was generated by a controlled evaporator mixing system (Bronkhorst, CEM) using Milli-Q ® water. The humidity in the flow tube was maintained constant at a mixing ratio of approximately 1.3 % except for measurements at the 95 temperature of 280 K, where the mixing ratio was reduced to 0.8 %, to prevent condensation. Ozone was generated by oxygen photolysis in a custom-built ozoniser, in which synthetic air was illuminated by the 185 nm radiation of a low-pressure mercury lamp. This resulted in ozone mixing ratios of 20 to 50 ppbv in the air mixture.

A quadrupled Nd:YAG laser (Lumibird, Ultra 100) providing short laser pulses at a wavelength of 266 nm with a high pulse energy of 20 mJ and a low repetition rate of 1 Hz was used to photolyse ozone producing excited oxygen atoms $O(^1D)$, which 100 subsequently reacted with water to form 2 OH radicals:

$$O_3 + h\nu \,(266 \, nm) \quad \rightarrow \quad O(^1D) + O_2 \tag{R1}$$

$$O(^1D) + H_2O \quad \rightarrow \quad 2\,OH \tag{R2}$$

Near the end of the flow tube, a small fraction of the air $(1 \, L min^{-1})$ was sampled through a conical nozzle into a low pressure cell (3.5 hPa). The OH radicals were excited by laser pulses at a wavelength of 308 nm generated by a custom-built 105 dye laser system operated at a high repetition rate of 8.5 kHz (Strotkamp et al., 2013). Perpendicular to the flow axis and the laser beam axis, the fluorescence of the OH radicals was detected by a multichannel plate detector (Photek, MCP 325) and single photons were counted by a multichannel scaler (Sigma Space, AMCS). The reaction time was determined by the



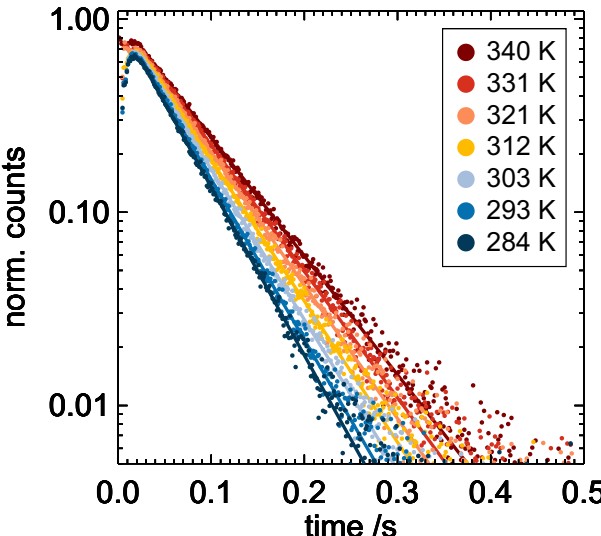

**Figure 2.** Measurement of the OH pseudo-first order rate coefficients at different temperatures (coloured points), when the air in the flow tube contained toluene with a mixing ratio of $115\,\mathrm{ppbv}$. Fluorescence counts are normalised to the amplitude derived from the single-exponential fit. No deviation from a single-exponential fit is observed for any of the conditions. Results of the fit are shown as coloured lines.

electronics and therefore did not depend on the flow rate of the gas in the reaction volume unlike in flow tube experiments using a sliding injector.

After the application of a photolysis laser pulse, the initial radical concentration, $[\mathrm{OH}]_0$, decayed in a pseudo-first order loss process with the rate $k$:

$$\mathrm{OH} = [\mathrm{OH}]_0 \exp(-k \cdot t) \tag{1}$$

The pseudo-first order rate coefficient of the OH decay was obtained by fitting the measured fluorescence counts to a single exponential function (including an offset) using a Levenberg-Marquardt minimisation procedure. The decay was observed for

$1\,\mathrm{s}$ with a time resolution of $1\,\mathrm{ms}$. Typically, 300 decay curves were summed up to improve the signal-to-noise ratio before the fit was applied. The measurement of OH by fluorescence is extremely sensitive, with a detection limit of the order of less than $1 \cdot 10^6\,\mathrm{cm}^{-3}$ at a time resolution of $1\,\mathrm{min}$, so that a high precision of the measured radical decay curve and the fitted rate coefficient was achieved.

     Radicals were also lost in diffusion limited wall reactions on the surface of the flow tube (Lou et al., 2010). This instrumental

zero-decay rate coefficient ($k_0$) was determined by measurements in pure synthetic air containing only $O_3$ and water vapour. The reaction rate constant ($k_{\mathrm{OH+X}}$) was then calculated from the measured OH reactivity, taking into account the zero-decay



rate coefficient $k_0$ and the reactant concentration [X] in the flow tube:

$$k_{\mathrm{OH+X}} = \frac{k - k_0}{[\mathrm{X}]} \tag{2}$$

To determine the temperature dependence of the rate coefficients, the OH reactant concentration was kept constant. The
reaction rate coefficients were measured at 7 different temperatures between 280 and 340 K (Fig.2) and the values are fitted to
the Arrhenius expression:

$$k(\mathrm{OH}, T) = A \exp\left(-\frac{E_A}{RT}\right) \tag{3}$$

where $E_A$ is the activation energy, $R$ is the gas constant, and $A$ is the pre-exponential factor. $E_A$ and $A$ can be calculated from
a linear fit of the logarithm of the rate coefficients against the reciprocal temperatures.

The initial radical concentration was approximately $3 \cdot 10^9 \, \mathrm{cm}^{-3}$ while the typical reactant concentrations were in excess,
making the reactions pseudo-first order. The results do not depend on the exact OH reactant concentration but an equivalent
OH reactivity significantly higher than the zero-decay rate coefficient should be used to minimise potential systematic errors
from the subtraction (Eq. 2).

The accuracy of the reaction rate constants was determined from the accuracy of OH reactant concentration and the repro-
ducibility of measurements. The potential error in the dilution is small as the accuracy of the measured flow rates from flow
controllers is better than $< 1\%$. If the OH reactant was mixed directly from a commercial gas standard, the accuracy of the
canister concentration is 1 to 2 %. If the canisters with OH reactants were prepared from liquids or diluted from gas standards
and the concentrations are measured by the total organic carbon (TOC) measurement (Section 2.2), the accuracy depends on
the purity of the liquid, which can be higher than 99 %, and on the measurement by the TOC method, which has an uncertainty
of 2 % (2-$\sigma$). The reproducibility of the measurements was 2 to 4 % (2-$\sigma$), which adds to the overall accuracy. Overall, a high
accuracy of 6 % (2-$\sigma$) for the OH rate coefficients was achieved except for methane, for which the accuracy was 12 % (2-$\sigma$)
due to the less accurate measurement of the methane concentration (8 %, 2-$\sigma$).

Deviations from a single-exponential decay of OH radicals in these measurements can only occur, if OH is formed from
products on the time scale of the measurement, as no additional reactants are present. This is possible for fast unimolecular
reactions of organic peroxy radicals (RO$_2$) such as one RO$_2$ isomer from the OH reaction of methacrolein, which can undergo
a 1,4 H-shift reaction at a rate of approximately $0.5 \, \mathrm{s}^{-1}$ at room temperature (Crounse et al., 2012; Fuchs et al., 2014). Since
the OH radical lifetime in the experiments in this work was at most $0.1 \, \mathrm{ms}$ and the precision of the measurement allowed to
observe the decay for $0.4 \, \mathrm{s}$ in this case, a deviation from a single-exponential decay would have been clearly observable.

## 2.2 Preparation of the OH reactants and determination of the reactant concentrations

The concentration of the OH reactant in the flow tube, where the reaction with OH radicals takes place, can be calculated from
the concentration of the prepared gas mixture, flow rates of flow controllers used for the dilution, and temperature and pressure
in the flow tube. The OH reactant concentrations in the flow tube were chosen to give an OH reactivity in the range of 10 to
$20 \, \mathrm{s}^{-1}$.



The 4 alkanes (methane, ethane, propane, n-butane) studied in this work were provided as high-purity gases (99.5 %, Linde),
which were diluted for the experiments. The other VOCs are liquids from which gas mixtures were prepared: Methyl vinyl ketone (MVK, purity 95.3 %, Thermo Scientific), $\gamma$-terpinene (purity 98.8 %, Sigma-Aldrich), $\Delta^3$-carene (purity 99.3 %, Sigma-Aldrich), myrcene (purity 92.1 %, SIAL), toluene (purity 100 %, Sigma-Aldrich), o-xylene (purity 99.6 %, Sigma-Aldrich),
m-xylene (purity 99.8 %, Thermo Scientific), mesitylene (purity 99.2 %, SIAL). The effect of impurities is not easily quantified as it is not clear, whether they are reactive hydrocarbons. For most of the species investigated in this work with purities
higher than > 95 %, it can be assumed that the possible effect of impurities on the results was less than a few percent.

The VOC gas mixtures were prepared in Restek SilcoCan canisters (6 L volume). The canister was first evacuated to a
pressure of less than 7 Pa. Either the canister was then filled with a small amount of the gaseous compound or the liquid VOC
was injected through a septum using a 10 μL syringe. In this case, a small flow of synthetic air flowed into the evacuated
canister behind the septum. The canister was then pressurised with synthetic air to approximately 350 kPa.

The concentration of the OH reactant (Eq. 2) was determined by measuring the total organic carbon (TOC) concentration in
the prepared canister. This was achieved using a high temperature catalytic oxidation of the reactant to carbon dioxide ($CO_2$).
This method has been successfully used, for example, to determine total organic carbon in particles (Stockwell et al., 2018;
Price et al., 2023).

In this method, a small flow of the OH reactant mixture in the SilcoNert ® coated canister (flow rate: few $cm^3/min$) is diluted
by a flow of synthetic air. Both flows are controlled by calibrated mass flow controllers. The air (flow rate: $\approx 400$ sccm, sccm:
$cm^3/min$ at standard conditions) is passed through a pre-oven at a temperature of 1033 K and then over a palladium catalyst
(Pd/MgAl2O4) at a temperature of 773 K. This can oxidise most organic species to $CO_2$ and water with few exceptions
(see below). The $CO_2$ concentration is measured using a cavity ring-down spectroscopy (CRDS) instrument (Picarro, G1301).
Assuming that all the carbon atoms are derived from the OH reactant, the mixing ratio of the organic compound in the canister
can be calculated from the number of carbons in each molecule and the dilution factor.

The TOC method was validated by measuring the concentration of a commercial propane gas mixture (Linde) in nitrogen
with a known mixing ratio of $(208 \pm 4)$ ppmv. The mixing ratio determined by the TOC method was $(210 \pm 2)$ ppmv, well
within the specifications of the gas cylinder.

The TOC method cannot be used for methane because methane cannot be fully oxidised under the operating conditions.
With the current catalyst design, ethane is also not fully converted when the mixing ratio is higher than 5 ppmv as noticed from
implausible results. This is probably due to the decrease in the $C-H$ bond energy with increasing number of carbons in the
homologue series of alkanes (Lide, 2004). Therefore, the ethane concentration was kept below 5 ppmv in the canister. Overall,
it can be assumed that the combustion was quantitative for all hydrocarbons for which the TOC method was used in agreement
with the results in Stockwell et al. (2018) and Price et al. (2023). The methane concentrations were measured using a cavity
ring-down spectrometer (Picarro G2401 CRDS). The Picarro CRDS instrument has a high precision (a few ppbv) and a high
linearity (0-20,000 ppmv) (Zellweger et al., 2012).





The myrcene used in this work was of low purity (92.1 %). This could introduce a systematic error in the concentration from the TOC measurement if it is assumed that all the carbon is derived from myrcene. This increases the uncertainty by 9 %, assuming that the impurity is all carbon and reduces the fraction of carbon dioxide derived from myrcene.

Three sets of experiments were performed for most species to determine the temperature dependence of the reaction rate constant. New VOC mixtures were prepared for each set of experiments.

### 2.3   Potential artefacts in the determination of OH rate coefficients

Since ozone is added to the gas mixture in the flow tube to produce the OH radicals (Reaction R1, R2), additional ozone reactions could play a role in the chemical system. However, the reactions of OH radicals with ozone are about two to four
orders of magnitude slower ($k_{OH+O_3} = 7.3 \cdot 10^{-14}\,\mathrm{cm^3 s^{-1}}$, $T = 298\,\mathrm{K}$, Burkholder et al. (2020)) than the studied reactions so that they do not play a role in the experiments.

     In addition, the studied unsaturated organic compounds could be degraded by ozonolysis in the flow tube, introducing a systematic error as the assumed concentrations in the calculation of the rate coefficient (Eq. 2) would be too high. However, since the ozone mixing ratio is low (20 to 30 ppbv) and the residence time in the flow tube is only a few seconds, the
consumption of the OH reactant by ozonolysis is negligible even for species with a high rate coefficient such as myrcene ($k_{myrc+O_3} = 4.7 \cdot 10^{-16}\,\mathrm{cm^3 s^{-1}}$, Grimsrud et al. (1975)).

     The ozonolysis of organic compounds can also lead to OH production, as the cycloaddition of ozone to the C=C double bond forms a primary ozonide, which dissociates into a stable carbonyl compound and a Criegee intermediate, which can undergo a decomposition reaction to form an OH radical (Criegee, 1975). This OH production could distort the single exponential OH
decay resulting from the pseudo-first order loss. Although OH production from ozonolysis reactions is fast with timescales of nanoseconds to milliseconds (Lester and Klippenstein, 2018; Novelli et al., 2014), this does not significantly affect the OH decay curve under the experimental conditions in this study. For example, for myrcene, the VOC with the highest ozonolysis rate coefficient investigated in this work, the integrated OH concentration from the ozone reaction is only 0.2 % of the initial OH concentration from the flash photolysis of ozone.

VOCs with more carbon double bonds may also be photolysed by the 266 nm radiation, which would affect the assumed concentration in Eq. 2. However, typical absorption cross sections for the compounds investigated in this work are in the range of $10^{-19}$ to $10^{-20}\,\mathrm{cm^2}$ at 266 nm (Śmiałek et al., 2012; Gierczak et al., 1997a; Kubala et al., 2009). Even if the quantum yield was 1, the fraction of photolysed VOCs in the flow tube would be less than 0.1 %.

## 3   Results and Discussion

### 3.1   Validation of temperature-dependent OH rate coefficients using alkanes

The temperature dependence of the reaction of alkanes with OH was determined in many studies and therefore this class of species was used to validate the method described in this work.





In some studies, the temperature behaviour is not described by the Arrhenius expression (Eq. 3) but in the form of a 3-parameter equation (Kooij equation) which takes into account the temperature dependence of the pre-exponential factor $A$ (Smith, 2008):

$$k(T) = A T^m \exp\left(\frac{E'_A}{RT}\right) \tag{4}$$

$m$ denotes an additional parameter. IUPAC provides Arrhenius expressions for rate constants that are optimised for a temperature range between $200\,\mathrm{K}$ and $300\,\mathrm{K}$. However, the studies, on which the recommendations are based, cover a much wider temperature range and IUPAC also provides a Kooij expression for the temperature dependence to describe the entire temperature range (Atkinson et al., 2006). The Arrhenius expressions from the NASA-JPL recommendation cover the entire temperature range tested in this work (Burkholder et al., 2020).

The reaction of the OH radical with the 4 alkanes ($C_nH_{2n+2}$ with $n = 1$ to 4) studied in this work proceeds via H-atom abstraction forming a water molecule and an alkyl radical which rapidly adds an oxygen molecule forming a peroxy radical:

$$OH + C_nH_{2n+2} \quad \rightarrow \quad H_2O + C_nH_{2n+1} \tag{R3}$$

$$C_nH_{2n+1} + O_2 + M \quad \rightarrow \quad C_nH_{2n+1}O_2 + M \tag{R4}$$

A single-exponential decay of the OH radicals is expected from this mechanism and was also observed for all OH decay curves in the experiments. The mixing ratios in the flow tube were $63\,\mathrm{ppmv}$ for methane, 1.4 to $1.8\,\mathrm{ppmv}$ for ethane, 0.4 to $0.6\,\mathrm{ppmv}$ for propane, and 0.2 to $0.3\,\mathrm{ppmv}$ for n-butane.

For methane (Table 1), the rate coefficients obtained in this work are in excellent agreement with measurements in all previous studies with maximum differences of a few percent (Atkinson et al., 2006). The parameters of the Arrhenius expressions derived in this study are also in a good agreement with all other studies within the uncertainties. In the study of Mellouki et al. (1994), the pre-exponential factor is 20 to $30\,\%$ higher than in this and other studies. Therefore, the results in this work confirm that this study should be excluded from the recommendations as done by IUPAC (Atkinson et al., 2006).

For ethane (Table 2), the rate coefficients in this study agree well with the results of the studies by Tully et al. (1986), Wallington et al. (1987), and Talukdar et al. (1994). The temperature dependence derived in all studies, including this work, gives similar results within $5\,\%$ over the temperature range tested in this work. Only the rate coefficients in the study by Tully et al. (1983), which is excluded in the IUPAC recommendations, are up to $15\,\%$ higher than in this work. Therefore, the parameters of the Arrhenius expression in this work are consistent with the parameterisation of the temperature dependence derived in other studies, e.g. the Kooij expression (Eq. 4) recommended by IUPAC (Atkinson et al., 2006) and the Arrhenius expression recommended by NASA-JPL (Burkholder et al., 2020) are in excellent agreement (better than $3\,\%$) with the expression in this work.

The Arrhenius expression for the rate coefficient of the OH reaction with propane (Table 3) is in perfect agreement (differences less than $3\,\%$) with the results of Mellouki et al. (1994) and Talukdar et al. (1994). The rate coefficients derived from the measurements in Droege and Tully (1986a) give a similar temperature dependence, but the values are approximately $7\,\%$ higher than in the other studies. In contrast, the temperature dependence derived in Kozlov et al. (2003) gives perfect agreement





**Table 1.** OH rate coefficients ($k$) of methane determined in this study and reported in the literature. All studies measured absolute rate coefficients. The IUPAC recommendations (Atkinson et al., 2006) for the temperature dependence include the Arrhenius expression optimised for the temperature range between 200 and 300 K, and recommended a Kooij expression (Eq. 4) from Gierczak et al. (1997b) for a wider range of temperatures. The error of the value from this work at 294 K is the 2-$\sigma$ accuracy. Errors of the Arrhenius expression are derived from the precision of the fit.

| $k$ (294 K) / $10^{-15}$ cm$^3$s$^{-1}$ | $A$ / cm$^3$s$^{-1}$ | $E_A/R$ / K | bath gas | temperature / K | pressure / hPa | reference |
|---|---|---|---|---|---|---|
| 5.8 | $1.59 \cdot 10^{-20}T^{2.84}$ | 978 | He/N$_2$ | 223–420 | 400 | Vaghjiani and Ravishankara (1991) |
| $6.3 \pm 3.1$ | $4.0 \cdot 10^{-12}$ | $1994 \pm 114$ | He | 223–420 | 1 | Finlayson-Pitts et al. (1992) |
| 5.7 | $9.65 \cdot 10^{-20}T^{2.58}$ | 1082 | He/N$_2$ | 223–420 | 400 | Dunlop and Tully (1993) |
| $6.3 \pm 3.1$ | $2.56 \cdot 10^{-12}$ | $1765 \pm 146$ | He | 223–420 | 130 | Mellouki et al. (1994) |
| $5.89 \pm 0.12$ | $1.85 \cdot 10^{-20}T^{2.82}$ | $987 \pm 6$ | He | 223–420 | 130 | Gierczak et al. (1997b) |
| $5.8 \pm 0.7$ | $0.57 \cdot 10^{-20}T^{3.01}$ | $959 \pm 36$ | He | 223–420 | 130 | Bonard et al. (2002) |
| $5.9 \pm 0.4$ | $1.85 \cdot 10^{-12}$ | $1690 \pm 100$ | N$_2$ | 200–300 | 1013 | IUPAC, Atkinson et al. (2006) |
| $5.9 \pm 0.6$ | $2.45 \cdot 10^{-12}$ | 1775 | air | 178–2025 | 1013 | NASA-JPL, Burkholder et al. (2020) |
| $6.1 \pm 1.0$ | $2.0 \cdot 10^{-12}$ | $1706 \pm 35$ | air | 280–340 | 1000 | this work |

**Table 2.** OH rate coefficients ($k$) of ethane determined in this study and reported in the literature. Absolute rate constants were measured in all studies. The IUPAC recommendations (Atkinson et al., 2006) for the temperature dependence include the Arrhenius expression optimised for the temperature range between 200 and 300 K and a Kooij expression (Eq. 4). The error of the value from this work at 294 K is the 2-$\sigma$ accuracy. Errors of the Arrhenius expression are derived from the precision of the fit.

| $k$ (294 K) / $10^{-13}$ cm$^3$s$^{-1}$ | $A$ / cm$^3$s$^{-1}$ | $E_A/R$ / K | bath gas | temperature / K | pressure / hPa | reference |
|---|---|---|---|---|---|---|
| 2.5 | $1.43 \cdot 10^{-14}T^{1.05}$ | 911 | Ar | 297–800 | 130 | Tully et al. (1983) |
| 2.40 | $8.51 \cdot 10^{-18}T^{2.06}$ | 430 | He | 293–705 | 800 | Tully et al. (1986) |
| $2.4 \pm 0.8$ | $8.4 \cdot 10^{-12}$ | $1050 \pm 100$ | Ar | 226–363 | 70 | Wallington et al. (1987) |
| $2.4 \pm 0.3$ | $10.3 \cdot 10^{-12}$ | $1108 \pm 40$ | He | 231–377 | 130 | Talukdar et al. (1994) |
| $2.3 \pm 0.2$ | $6.9 \cdot 10^{-12}$ | $1000 \pm 100$ | N$_2$ | 200–300 | 1013 | IUPAC, Atkinson et al. (2006) |
| 2.36 | $1.49 \cdot 10^{-17}T^{2}$ | 499 | N$_2$ | 180–1230 | 1013 | IUPAC, Atkinson et al. (2006) |
| $2.39 \pm 0.12$ | $7.66 \cdot 10^{-12}$ | 1020 | air | 226–2000 | 1013 | NASA-JPL, Burkholder et al. (2020) |
| $2.34 \pm 0.12$ | $7.9 \cdot 10^{-12}$ | $1031 \pm 19$ | air | 280–340 | 1000 | this work |





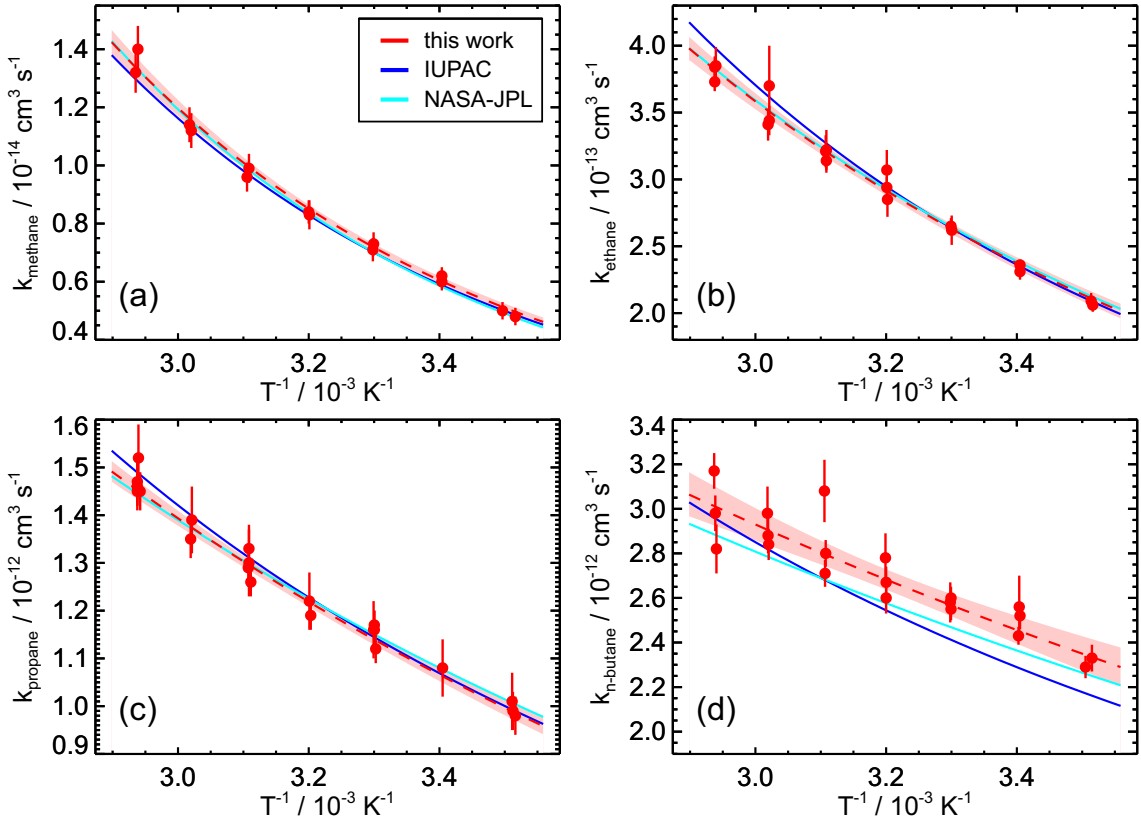

**Figure 3.** Temperature dependent rate coefficients of the OH reaction with four alkanes (a) methane, (b) ethane, (c) propane and (d) n-butane. The data points are fitted to an Arrhenius expression (dashed line) and compared with the 3-parameter expression of the IUPAC (Atkinson et al., 2006) and the Arrhenius expression of the NASA-JPL (Burkholder et al., 2020) recommendation. The coloured area indicates the 95 % confidence interval of the fit.

for the rate coefficient at room temperature, but higher values are obtained at higher temperatures (maximum difference of 7 % at 345 K).

The IUPAC recommendations (Atkinson et al., 2006) also include measurements by Clarke et al. (1998) and Donahue et al. (1998), in which the rate coefficient was measured at different temperatures, but no Arrhenius expression was determined. The rate coefficients obtained in these studies are in excellent agreement with those obtained in this work. The IUPAC recommended Arrhenius expression tends to give lower values than in these studies at higher temperatures (7 % at 345 K), but reproduces the value at room temperature well, as the fit is not optimised for temperatures above 300 K. However, the IUPAC Kooij expression (Eq. 4) describes the temperature behaviour observed in this study even better. The Arrhenius expression recommended by NASA-JPL (Burkholder et al., 2020) covers the entire temperature range and is in excellent agreement with the results of this work (maximum difference with the results of this work: 2 %).





**Table 3.** OH rate coefficients ($k$) of propane determined in this study and reported in the literature. Absolute rate constants were measured in all studies. The IUPAC recommendations (Atkinson et al., 2006) for the temperature dependence include the Arrhenius expression optimised for the temperature range between 200 and 300 K and a Kooij expression (Eq. 4). The error of the value from this work at 304 K is the 2-$\sigma$ accuracy. Errors of the Arrhenius expression are derived from the precision of the fit.

| $k$ (304 K) / $10^{-12}$ cm$^3$s$^{-1}$ | $A$ / cm$^3$s$^{-1}$ | $E_A/R$ / K | bath gas | temperature / K | pressure / hPa | reference |
|---|---|---|---|---|---|---|
| 1.20 | $1.04 \cdot 10^{-16}T^{1.72}$ | 145 | He | 293–854 | 530 | Droege and Tully (1986a) |
| $1.16 \pm 0.11$ | $0.98 \cdot 10^{-11}$ | $650 \pm 30$ | He | 233–363 | 130 | Mellouki et al. (1994) |
| $1.16 \pm 0.18$ | $1.01 \cdot 10^{-11}$ | $657 \pm 46$ | He | 233–376 | 130 | Talukdar et al. (1994) |
| 1.17 | $5.81 \cdot 10^{-17}T^{1.83}$ | 167 | Ar | 210–480 | 40 | Kozlov et al. (2003) |
| $1.11 \pm 0.09$ | $0.76 \cdot 10^{-11}$ | $585 \pm 100$ | N$_2$ | 200–300 | 1013 | IUPAC, Atkinson et al. (2006) |
| 1.15 | $1.65 \cdot 10^{-17}T^{2}$ | 85 | N$_2$ | 190–1220 | 1013 | Atkinson et al. (2006) |
| $1.16 \pm 0.03$ | $0.92 \cdot 10^{-11}$ | 630 | air | 190–908 | 1013 | NASA-JPL,Burkholder et al. (2020) |
| $1.15 \pm 0.1$ | $1.04 \cdot 10^{-11}$ | $670 \pm 23$ | air | 280–340 | 1000 | this work |

**Table 4.** OH rate coefficients ($k$) of butane determined in this study and reported in the literature. Absolute rate constants were measured in all studies. The IUPAC recommendations (Atkinson et al., 2006) for the temperature dependence include the Arrhenius expression optimised for the temperature range between 200 and 300 K and a Kooij expression (Eq. 4). The error of the value from this work at 294 K is the 2-$\sigma$ accuracy. Errors of the Arrhenius expression are derived from the precision of the fit.

| $k$ (294 K) / $10^{-12}$ cm$^3$s$^{-1}$ | $A$ / cm$^3$s$^{-1}$ | $E_A/R$ / K | bath gas | temperature / K | pressure / hPa | reference |
|---|---|---|---|---|---|---|
| $2.4 \pm 0.8$ | $1.41 \cdot 10^{-11}$ | $524 \pm 93$ | | 298–495 | 40 | Greiner (1970) |
| $2.6 \pm 1.4$ | $1.76 \cdot 10^{-11}$ | $559 \pm 151$ | Ar | 298–420 | 40 | Perry et al. (2008) |
| 2.40 | $2.34 \cdot 10^{-17}T^{1.95}$ | −134 | He | 294–509 | 530 | Droege and Tully (1986b) |
| $2.35 \pm 0.06$ | $2.04 \cdot 10^{-17}T^{2}$ | $-(85 \pm 8)$ | He | 231–378 | 130 | Talukdar et al. (1994) |
| $2.31 \pm 0.14$ | $0.98 \cdot 10^{-11}$ | $425 \pm 100$ | N$_2$ | 200–300 | 1013 | IUPAC, Atkinson et al. (2006) |
| 2.29 | $2.03 \cdot 10^{-17}T^{2}$ | −78 | N$_2$ | 185–509 | 1013 | IUPAC, Atkinson et al. (2006) |
| $2.36 \pm 0.07$ | $1.02 \cdot 10^{-11}$ | 430 | air | 185–509 | 1013 | NASA-JPL, Burkholder et al. (2020) |
| $2.5 \pm 0.16$ | $1.1 \cdot 10^{-11}$ | $441 \pm 35$ | air | 280–340 | 1000 | this work |



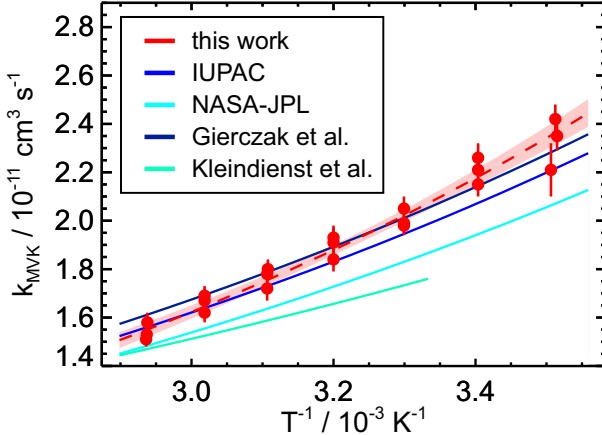

**Figure 4.** Temperature dependence of rate coefficients of the reaction of OH with methyl vinyl ketone (MVK). The data points measured in this work at different temperatures are fitted to an Arrhenius expression (dashed line) and compared with literature values. The coloured area indicates the 95 % confidence interval of the fit.

The results of this study for the temperature dependence of the rate coefficient of the OH reaction with n-butane (Table 4) agree best with the results of Droege and Tully (1986b) and Talukdar et al. (1994). The rate coefficients of Perry et al. (2008) show a similar temperature dependence, but the values are up to 15 % higher than in the other studies. Overall, the variability of the values obtained at room temperature in the different studies is higher than for the other alkanes (Atkinson et al., 2006). The parameterisation of the temperature dependence of the rate coefficient of the IUPAC (Atkinson et al., 2006) and NASA-JPL (Burkholder et al., 2020) recommendations are well within the uncertainties of the measurements in this work.

In summary, the excellent agreement between the temperature dependence of the reaction rate constants for alkanes determined in this work and the recommendations in the literature validates the methods used in this work and provides confidence that results for other species for which literature values are not available or have larger uncertainties are reliable.

## 3.2 Reaction of OH with methyl vinyl ketone (MVK)

Methyl vinyl ketone (MVK, but-3-en-2-one) is an oxidation product of isoprene and contains a carbon double bond, to which OH adds.

Measurements of the rate coefficients for the reaction of OH with MVK at temperatures between 280 and 340 K were fitted to an Arrhenius expression (Fig. 4), resulting in a rate coefficient of

$$k_{\mathrm{OH+MVK}} = 1.80 \cdot 10^{-12} \exp\left((733 \pm 27)\,\mathrm{K} \cdot T^{-1}\right)\, cm^3 s^{-1} \tag{5}$$

This expression describes the data well with a maximum difference between individual measurements and fitted values of 6 %. The rate coefficients show a negative temperature dependence, as can be seen from the temperature dependence of the $E_A/R$ coefficient.





**Table 5.** Rate coefficients of OH with MVK determined in this study and reported in the literature for at a certain pressure ($p$) and temperature ($T$). The error of the value from this work at 294 K is the 2-$\sigma$ accuracy. Errors of the Arrhenius expression are derived from the precision of the fit.

| $k_{\mathrm{MVK+OH}}$ (294 K) / $10^{-11}\,\mathrm{cm^3s^{-1}}$ | $A$ / $10^{-12}\,\mathrm{cm^3s^{-1}}$ | $E_A/R$ / K | $T$ / K | $p$ / hPa | bath gas | method | reference |
|---|---|---|---|---|---|---|---|
| $1.8 \pm 0.4$ | 3.85 | $-456 \pm 73$ | 298-424 | 70 | Ar | FP-RF[a] | Kleindienst et al. (1982) |
| $2.1 \pm 0.4$ | 2.67 | $-612 \pm 49$ | 232-378 | 130 | He | LP-LIF[b] | Gierczak et al. (1997a) |
| $2.1 \pm 1.4$ | 2.6 | $-610 \pm 200$ | 230-380 | 1013 | $N_2$ | | IUPAC, Mellouki et al. (2021) |
| $1.94 \pm 0.15$ | 2.7 | $-580$ | 232-424 | 1013 | air | | NASA-JPL, Burkholder et al. (2020) |
| $2.21 \pm 0.2^c$ | 1.80 | $-733 \pm 27$ | 280-340 | 1000 | air | LP-LIF[b] | this work |

[a]flash photolysis resonance fluorescence, [b]laser flash photolysis laser-induced fluorescence, [c]calculated from the Arrhenius expression (Eq. 3)

To date, the temperature dependence of the reaction with MVK has only been investigated in two other studies using absolute rate methods (Table 5, Kleindienst et al. (1982); Gierczak et al. (1997a)). The rate constant at 298 K preferred by the IUPAC recommendation is based on three room-temperature studies. The temperature dependence is taken from the study by Gierczak et al. (1997a) because a wide temperature range was covered in that study. The Arrhenius fit obtained in this work is in good agreement with the results of Gierczak et al. (1997a) and therefore also with the IUPAC recommendation (e.g., 4 % deviation at $T = 298$ K). In Kleindienst et al. (1982) only few temperatures were tested, and the data are scattered. This explains the differences between the Arrhenius parameters obtained in their study and the results of this study and from Gierczak et al. (1997a). As the study by Kleindienst et al. (1982) is included in the NASA-JPL recommendation, the Arrhenius expression in the NASA-JPL recommendation also gives slightly lower values.

### 3.3 Reactions of OH with monoterpenes ($\Delta^3$-carene / $\gamma$-terpinene / myrcene)

The rate coefficients of OH with the three monoterpenes, $\Delta^3$-carene, $\gamma$-terpinene and myrcene, were measured in the temperature range of 280 K to 340 K. The temperature dependence of the reaction kinetics of OH + sabinene using the method in this work was described in Pang et al. (2023). The reaction proceeds predominantly by OH addition to the carbon double bonds present in the monoterpenes. The data were fitted to the Arrhenius expression (Table 6), giving in reaction rate coefficients of

$$k_{\mathrm{OH}+\Delta^3-\mathrm{carene}} = 2.8 \cdot 10^{-11} \exp\left((321 \pm 40)\mathrm{K} \cdot T^{-1}\right) cm^3 s^{-1} \tag{6}$$

$$k_{\mathrm{OH}+\gamma-\mathrm{terpinene}} = 7.5 \cdot 10^{-12} \exp\left((915 \pm 38)\mathrm{K} \cdot T^{-1}\right) cm^3 s^{-1} \tag{7}$$

$$k_{\mathrm{OH}+\mathrm{myrcene}} = 1.05 \cdot 10^{-11} \exp\left((833 \pm 28)\mathrm{K} \cdot T^{-1}\right) cm^3 s^{-1} \tag{8}$$

The residuals of the fits are within 6 % for all three species (Fig. 5).

The rate coefficients of all 3 species show a negative temperature behaviour (Fig. 5). While the values of the $E_A/R$ coefficients are similar for $\gamma$-terpinene and myrcene, they are significantly lower for $\Delta^3$-carene (($-321 \pm 40$) K). This is due to




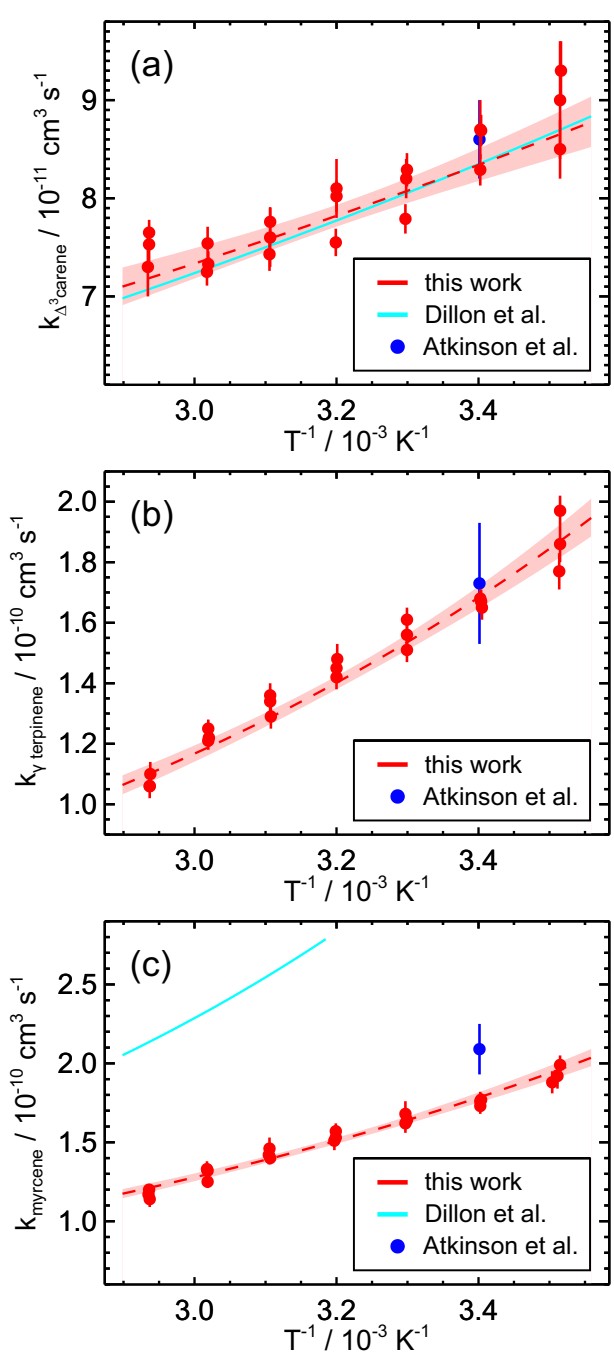

**Figure 5.** Temperature dependence of the rate coefficients of the reaction of OH with a) $\Delta^3$-carene b) $\gamma$-terpinene and c) myrcene determined in this work (Arrhenius fit: dashed line) and reported in the literature. The data points measured in this work at different temperatures are fitted to an Arrhenius expression. The coloured area indicates the 95 % confidence interval of the fit.



**Table 6.** Rate coefficients of OH with $\Delta^3$-carene, $\gamma$-terpinene and myrcene determined in this study and reported in the literature for at a certain pressure ($p$) and temperature ($T$). The error of the values from this work is derived from the 2-$\sigma$ accuracy of 3 % and the reproducibility of the measurement. Literature values are calculated for the temperature of 294 K if a temperature dependent expression is available. The error of the value from this work at 294 K is the 2-$\sigma$ accuracy. Errors of the Arrhenius expression are derived from the precision of the fit.

| $k_{\text{VOC+OH}}$ / $10^{-10}\,\text{cm}^3\text{s}^{-1}$ | $A$ / $10^{-11}\,\text{cm}^3\text{s}^{-1}$ | $E_A/R$ / K | $T$ / K | $p$ / hPa | bath gas | method | reference |
|---|---|---|---|---|---|---|---|
| $\Delta^3$-carene | | | | | | | |
| $0.86 \pm 0.04$ | – | – | 294 | 980 | air | GC-FID[a] | Atkinson et al. (1986) |
| $0.84 \pm 0.11$ | 2.48 | $-357 \pm 17$ | 235-357 | 266 | $N_2$/air | LP-LIF[b] | Dillon et al. (2017) |
| $0.83 \pm 0.06$ | 2.8 | $-321 \pm 40$ | 280-340 | 1000 | air | LP-LIF[b] | this work |
| $\gamma$-terpinene | | | | | | | |
| $1.73 \pm 0.2$ | – | – | 294 | 980 | air | GC-FID[b] | Atkinson et al. (1986) |
| $1.67 \pm 0.2$ | 0.75 | $-915 \pm 38$ | 280-340 | 1000 | air | LP-LIF[b] | this work |
| myrcene | | | | | | | |
| $2.09 \pm 0.16$ | – | – | 294 | 980 | air | GC-FID[a] | Atkinson et al. (1986) |
| $3.5 \pm 1.0$ | 0.92 | $-1071 \pm 82$ | 313-423 | 1000 | He | MS[c] | Hites and Turner (2009) |
| $1.8 \pm 0.3$ | 1.05 | $-833 \pm 28$ | 280-340 | 1000 | air | LP-LIF[b] | this work |

[a] gas chromatography – flame ionization detector, reference species: 2,3-dimethyl-2-butene, [b] laser flash photolysis laser-induced fluorescence, [c] mass spectrometry, reference species: isoprene

the structural difference between the species. Monoterpenes with a bicyclic structure such as $\Delta^3$-carene and $\alpha$-pinene show a
similar temperature dependence of the OH coefficient ($\alpha$-pinene: $E_A/R = (-330 \pm 6)$ K, Dillon et al. (2017)). Monoterpene species containing an additional exocyclic carbon double bond such as sabinene ($E_A/R = (-537 \pm 30)$ K, Pang et al. (2023)) and $\beta$-pinene ($E_A/R = (-460 \pm 150)$ K, Mellouki et al. (2021)) have slightly higher coefficients like obtained in this work for similar species.

The Arrhenius expression for the reaction of OH with $\Delta^3$-carene determined in this work shows excellent agreement (e.g.,
1 % deviation at 298 K) with a recent absolute rate study by Dillon et al. (2017), on which the IUPAC recommendation is also based. The uncertainty of the fit parameters in this study is in the same range as that of the IUPAC recommendation (12 %). There is only one other study by Atkinson et al. (1986), in which the rate coefficient was determined at room temperature ($T = 294$ K) using a relative rate method (Table 6), which is in good agreement with the results of the other studies at this temperature.

For $\gamma$-terpinene, the temperature dependence of the OH rate coefficient has not yet been investigated. The uncertainty of the Arrhenius expression obtained in this work is 18 %. The only literature value available is from a kinetic study at a single



temperature ($T = 294$ K, Atkinson et al. (1986)), on which also the IUPAC recommendation is based on. The results obtained in this work are in excellent agreement with the result of Atkinson et al. (1986) (e.g., 2 % deviation at $T = 294$ K).

There are only 2 relative rate studies (Table 6) in the literature investigating the rate coefficient of OH with myrcene, one
at $T = 294$ K (Atkinson et al., 1986) and the other over the temperature range of 313 and 423 K (Hites and Turner, 2009). IUPAC only gives a recommendation for room temperature ($T = 298$ K) based on the study by Atkinson et al. (1986). The rate coefficients calculated from the Arrhenius expression in this work, with a maximum uncertainty of 17 %, are in good agreement with the value from Atkinson et al. (1986) (18 % deviation).

Extrapolation of the Arrhenius expression by Hites and Turner (2009) to room temperature gives a rate coefficient about
320 50 % higher than that measured by Atkinson et al. (1986) (Table 6). Their Arrhenius expression also gives significantly higher values than those derived from the expression in this work (e.g., 88 % deviation at $T = 313$ K). As discussed in Hites and Turner (2009), one reason for the higher values in their work could be due to a bias in the measured concentrations of organic compounds as they were detected by mass spectrometry, which could give too high concentrations if several species with the same mass-to-charge ratio are detected.

**3.4 Reactions of OH with aromatic hydrocarbons (toluene / mesitylene / o-xylene / m-xylene)**

The temperature dependence of the rate coefficient of OH with four aromatic hydrocarbons, toluene (methylbenzene), mesitylene (1,3,5-trimethylbenzene), m-xylene (1,3 dimethylbenzene), and o-xylene (1,2 dimethylbenzene) was investigated (Fig. 6). All data were fitted to the Arrhenius expression in the temperature range of 280 to 340 K giving a maximum residual of 4 to 6 %. All rate coefficients show a negative temperature behaviour. The reactions of the aromatic hydrocarbons with OH can
proceed via different reaction pathways: hydrogen abstraction from the aromatic ring or from a substituent, or OH addition to the aromatic ring. At temperatures below $T = 325$ K, the electrophilic ring addition is the dominant pathway (Atkinson and Aschmann, 1989).

With respect to the OH reaction with toluene, the OH-toluene adduct is thermally unstable and can decompose prior to the addition of oxygen to form a peroxy radical (Vereecken, 2019). This could lead to a non-exponential decay of the OH, as OH
could be reformed on the time scale of the observed OH decay (Tully et al., 1981). In this work, only single exponential OH decays were observed in the presence of oxygen and at ambient pressure. In this case, the lifetime of the adduct with respect to its re-dissociation is of the order of seconds, much longer than the lifetime with respect to the oxygen addition leading to the formation of a peroxy radical, which is of the order of milliseconds (Bohn and Zetzsch, 1999; Alarcon et al., 2015), so that the latter reaction dominates.

The data from the measurements were fitted to the Arrhenius expression (Table 7), resulting in rate coefficients of

$$k_{\text{OH+toluene}} = 1.39 \cdot 10^{-12} \exp\left((430 \pm 32)\text{K} \cdot T^{-1}\right) \, cm^3 s^{-1} \tag{9}$$

$$k_{\text{OH+mesitylene}} = 3.3 \cdot 10^{-12} \exp\left((861 \pm 28)\text{K} \cdot T^{-1}\right) \, cm^3 s^{-1} \tag{10}$$

$$k_{\text{OH+m-xylene}} = 1.38 \cdot 10^{-12} \exp\left((861 \pm 31)\text{K} \cdot T^{-1}\right) \, cm^3 s^{-1} \tag{11}$$

$$k_{\text{OH+o-xylene}} = 1.86 \cdot 10^{-12} \exp\left((595 \pm 29)\text{K} \cdot T^{-1}\right) \, cm^3 s^{-1} \tag{12}$$



**Table 7.** Rate coefficient of OH with the four aromatic hydrocarbons toluene, mesitylene, o-xylene and m-xylene determined in this study and reported in the literature for at a certain pressure ($p$) and temperature ($T$). Literature values are calculated for the temperature of 294 K if a temperature dependent expression is available. The error of the value from this work at 294 K is the 2-$\sigma$ accuracy. Errors of the Arrhenius expression are derived from the precision of the fit.

| $k_{\text{VOC+OH}}$ / $10^{-11}$ cm$^3$s$^{-1}$ | $A$ / cm$^3$s$^{-1}$ | $E_A/R$ / K | $T$ / K | $p$ / hPa | bath gas | method | reference |
|---|---|---|---|---|---|---|---|
| **toluene** | | | | | | | |
| $0.88 \pm 0.03$ | $0.79 \cdot 10^{-12}$ | $-614 \pm 114$ | 284-363 | 970 | air | GC-FID[a,b] | Semadeni et al. (1995) |
| $0.70 \pm 0.07$ | $3.8 \cdot 10^{-12}$ | $-180 \pm 170$ | 213-298 | 130 | Ar | FP-RF[c] | Tully et al. (1981) |
| $0.57 \pm 0.08$ | $1.8 \cdot 10^{-12}$ | $-340 \pm 200$ | 210-350 | 1013 | N$_2$ | | IUPAC, Mellouki et al. (2021) |
| $0.60 \pm 0.04$ | $1.39 \cdot 10^{-12}$ | $-430 \pm 32$ | 280-340 | 1000 | air | LP-LIF[e] | this work |
| **mesitylene** | | | | | | | |
| $5.7 \pm 0.5$ | – | – | 296 | 1010 | air | GC-FID[a,f] | Kramp and Paulson (1998) |
| $5.42 \pm 0.11$ | $0.44 \cdot 10^{-11}$ | $-738 \pm 176$ | 278-347 | 980 | air | GC-FID[a,g] | Aschmann et al. (2006) |
| $6.1 \pm 0.2$ | $1.32 \cdot 10^{-11}$ | $-450 \pm 50$ | 275-340 | 380/750 | He | FP-RF[c] | Bohn and Zetzsch (2012) |
| $7.1 \pm 0.9$ | $(1.1 \pm 0.6) \cdot 10^{-11}$ | $-550 \pm 180$ | 299-348 | 200 | He | FP-RF[c] | Alarcon et al. (2015) |
| $6.0 \pm 0.6$ | $0.33 \cdot 10^{-11}$ | $-861 \pm 28$ | 280-340 | 1000 | air | LP-LIF[d] | this work |
| **m-xylene** | | | | | | | |
| $2.3 \pm 0.4$ | – | – | 296 | 980 | air | GC-FID[a,g] | Atkinson and Aschmann (1989) |
| $2.14 \pm 0.14$ | – | – | 298 | 1 | He | DF-MS[h] | Mehta et al. (2009) |
| $2.4 \pm 0.3$ | – | – | 298 | 130 | Ar | FP-RF[c] | Hansen et al. (1975) |
| $2.4 \pm 0.3$ | – | – | 298 | 130 | Ar | FP-RF[c] | Perry et al. (1977) |
| $2.03 \pm 0.19$ | – | – | 298 | 266 | Ar | FP-RF[c] | Ravishankara et al. (1978) |
| $2.54 \pm 0.35$ | $(68 \pm 23) \cdot 10^{-12}$ | $1540 \pm 240$ | 500-1000 | 130 | Ar/He | FP-RF[c] | Nicovich et al. (1981) |
| $2.5 \pm 0.2$ | $1.38 \cdot 10^{-12}$ | $-861 \pm 31$ | 280-340 | 1000 | air | LP-LIF[d] | this work |
| **o-xylene** | | | | | | | |
| $1.22 \pm 0.19$ | – | – | 296 | 980 | air | GC-FID[a,g] | Atkinson and Aschmann (1989) |
| $1.19 \pm 0.08$ | – | – | 298 | 1 | He | DF-MS[h] | Mehta et al. (2009) |
| $1.53 \pm 0.15$ | – | – | 298 | 130 | Ar | FP-RF[c] | Hansen et al. (1975) |
| $1.43 \pm 0.15$ | – | – | 298 | 130 | Ar | FP-RF[c] | Perry et al. (1977) |
| $1.24 \pm 0.01$ | – | – | 298 | 266 | Ar | FP-RF[c] | Ravishankara et al. (1978) |
| $1.42 \pm 0.17$ | $(65 \pm 11) \cdot 10^{-12}$ | $1520 \pm 120$ | 500-1000 | 130 | Ar/He | FP-RF[c] | Nicovich et al. (1981) |
| $1.37 \pm 0.12$ | $1.86 \cdot 10^{-12}$ | $-595 \pm 29$ | 280-340 | 1000 | air | LP-LIF[d] | this work |

[a]gas chromatography – flame ionization detector, [b]reference species: 2,3-dimethyl butane, [c]flash photolysis – resonance fluorescence, [d]laser flash photolysis laser-induced fluorescence, [e]ten different reference substances, [f]reference species: $\alpha$-pinene, [g]reference species: propene, [h]discharge flow – mass spectrometry, reference substance: 1,4-dioxane





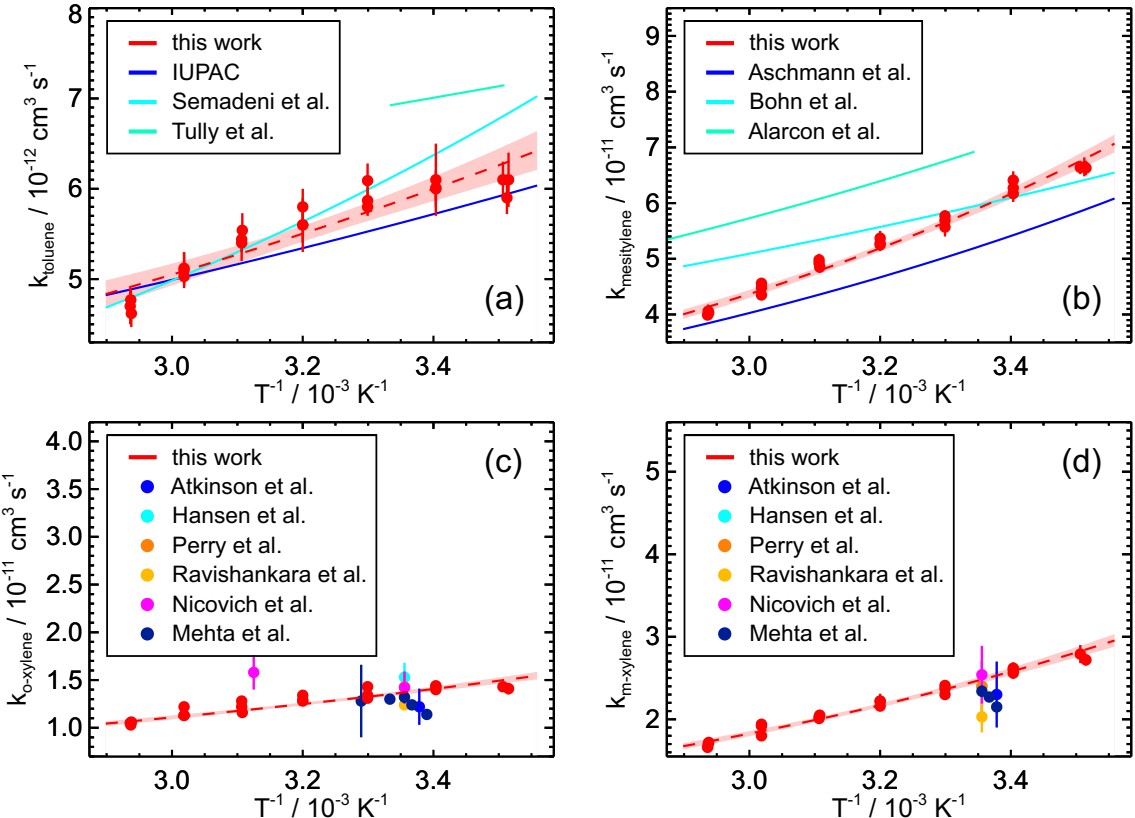

**Figure 6.** Temperature dependence of the rate coefficients of the reaction of OH with a) toluene b) mesitylene, c) o-xylene and d) m-xylene. The data points measured in this work at different temperatures are fitted to an Arrhenius expression (dashed lines) and compared with parameterisations reported in the literature, where available. The coloured area indicates the 95 % confidence interval of the fit. For o-xylene and m-xylene, no Arrhenius expression is reported in the literature, but values measured at single temperatures are shown.

Like for the other species, the differences between the Arrhenius expression and the measured values are in the range of a few percent. The temperature dependence of the reaction of OH with toluene has been investigated in several studies using relative and absolute rate methods, respectively (Perry et al., 1977; Tully et al., 1981; Knispel et al., 1990; Semadeni et al., 1995). An Arrhenius expression was derived only in the studies by Tully et al. (1981) (Table 7). These studies were also included in the IUPAC recommended value for the temperature range below 350 K, where the OH addition reaction dominates.

The Arrhenius expression derived in this work is in excellent agreement with the IUPAC recommendation for temperatures below 350 K (e.g., 4 % deviation at 298 K) and is therefore also in agreement with the results of the studies considered by IUPAC. However, the uncertainty of the Arrhenius expression derived in this work is significantly smaller (e.g., 16 % at $T = 298$ K) than the uncertainties given for the IUPAC recommendation (67 %) and the results in the studies by Semadeni et al. (1995) (57 %) and Tully et al. (1981) (38 %).



As in the experiments with toluene, no deviation from a single-exponential behaviour was observed in the measured OH decay curves for the conditions in the experiments with mesitylene because the consecutive reaction of the mesitylene-OH adduct with $O_2$ dominates, so that the dissociation of the releasing OH radicals cannot compete (Knispel et al., 1990; Koch et al., 2007). In contrast, a complex shape of the OH decays was observed in the experiments of Bohn and Zetzsch (2012) and Alarcon et al. (2015), who measured the rate coefficients using a similar method like in this study. They concluded that

for the mesitylene-OH adduct two different ortho- and ipso-types with respect to the methyl groups have to be considered. Differences with the observations in this work are explained by the different experimental conditions in Bohn and Zetzsch (2012) and Alarcon et al. (2015), where experiments were performed at reduced pressure and in the absence of oxygen as helium was used as bath gas.

    The temperature dependence of the reaction of OH with mesitylene was investigated by Aschmann et al. (2006), Bohn and

Zetzsch (2012) and Alarcon et al. (2015), but no recommendation is given by IUPAC (Table 7). The Arrhenius expression obtained in this study lies in between those reported in the literature, with a temperature dependence similar to that found by Aschmann et al. (2006) and Alarcon et al. (2015). However, the uncertainty of the Arrhenius fit parameters derived in this work is much smaller (e.g., 12 % at 298 K) than the uncertainty of those of Aschmann et al. (2006) (59 %) and Alarcon et al. (2015) (60 %).

The temperature dependence of the reactions of OH with the two isomers m-xylene and o-xylene was investigated for the first time, while the rate coefficient around room temperature ($T = 298$ K) was determined in several studies. Measurements using absolute rate methods were performed at low pressures of about 130 and 260 hPa using a noble gas as a bath gas (Table 7, Hansen et al. (1975); Perry et al. (1977); Ravishankara et al. (1978); Nicovich et al. (1981)). Relative rate studies have mostly been performed at ambient pressure (Doyle et al., 1975; Cox et al., 1980; Atkinson et al., 1983; Ohta and Ohyama, 1985;

Edney et al., 1986; Atkinson and Aschmann, 1989; Aschmann and Atkinson, 1998; Kramp and Paulson, 1998; Anderson et al., 2003; Mehta et al., 2009; Han et al., 2018). The rate coefficients measured in this work are generally in good agreement with the other studies in the temperature range of $T = 310$ and 280 K (Table 7).

    In the study by Mehta et al. (2009), the rate coefficients were also determined at temperatures of 320 and 340 K and below 280 K and various pressure values, but no Arrhenius expression was derived. The largest discrepancy (15 %) with the results

in this work is found for the highest temperature ($T = 340$ K) and low pressure ($p = 10$ hPa, not shown in Fig. 6). Nicovich et al. (1981) studied the temperature dependence in the range between 500 and 1000 K, but this is not relevant for processes in the troposphere.

    Han et al. (2018) calculated an Arrhenius expression $k_{\mathrm{OH+o-xylene}} = 6.24 \cdot 10^{-12} \exp\left((203 \pm 126)\mathrm{K} \cdot T^{-1}\right) \mathrm{cm}^3 \mathrm{s}^{-1}$ for the temperature dependence of the OH reaction with o-xylene. This parametrisation has a less strong temperature dependence

of the $E_A/R$ coefficient than that determined in this work (Table 7). The data used by Han et al. (2018) mainly include measurements made around room temperature, which results in a high uncertainty of 62 % for the $E_A/R$ parameter. The Arrhenius expression derived in this work covers a wider temperature range and an absolute rate technique is used, resulting in a low uncertainty of 14 %.



## 4 Conclusions

This study focused on the investigation of the temperature dependence of the rate coefficients for the gas-phase reaction between OH radicals and selected volatile organic compounds (VOCs), using OH reactivity measurements (laser-flash photolysis combined with the OH detection by laser-induced fluorescence, LP-LIF) and the total organic carbon (TOC) method to determine the initial VOC concentration. The OH reactivity instrument consists of a flow tube, in which the decay of the OH radicals is directly observed with a high sensitivity as they react with an organic compound. If the concentration of the OH

reactant is known and present in excess, the rate coefficient can be calculated directly by fitting the OH decay curve to a single-exponential function. A deviation from a single-exponential OH decay, as often observed for example in similar studies for aromatic hydrocarbons reported in the literature, could not be observed in this work. This can be explained by the experimental conditions used in this study ($p = 100\,\mathrm{hPa}$, $T = 280$ to $340\,\mathrm{K}$, in humidified air) and the low concentrations of OH radicals and the OH reactants, which reduced the potential effects of secondary chemistry.

The repetition of the experiments, including the measurement of the OH reactivity, the canister preparation, and the determination of the VOC concentration showed a high reproducibility with an average scatter of only 1.6 % (1-$\sigma$), indicating accurate measurements.

The high accuracy is demonstrated by the excellent agreement of the well studied rate coefficients of methane, ethane, propane and n-butane with IUPAC and NASA-JPL recommendations. The deviations of the values at room temperature are

405 less than 6 %. The rate coefficients determined for methyl vinyl ketone (MVK), $\Delta^3$-carene, toluene and mesitylene are also in good agreement with those reported in the literature, with maximum deviations of less than 7 %.

The results for myrcene do not agree well with a kinetic study investigating the temperature dependence over a narrow temperature range (Hites and Turner, 2009). However, the result obtained in this work shows a much better agreement with the IUPAC recommended value for room temperature ($T = 298\,\mathrm{K}$). For aromatic compounds no deviations from a single-

410 exponential OH decay were observed because the presence of oxygen minimises the dissociation of the VOC-OH adduct and subsequent OH release which would cause a deviation from a single-exponential OH decay.

To the best of our knowledge, this is the first time that the temperature dependence of the reactions between OH and $\gamma$-terpinene has been investigated in an atmospheric relevant temperature range. The values of the rate coefficients reported in the literature for single temperatures largely overlap with the values obtained in this work, within their uncertainties.

Overall, this work contributes to improving the database of atmospheric reaction kinetics for several atmospherically relevant hydrocarbons for which little data are available. As the OH reactivity measurement provides highly accurate and precise first-order rate coefficients, the accuracy of the values is mainly limited by the uncertainty of the hydrocarbon concentration in the measurements. This is minimised by the measurement of total organic carbon, which relies mainly on the assumption that all carbon that is detected as $CO_2$ can be attributed to the hydrocarbon.

*Data availability.* The data is listed in Tables in the Supplement.



*Author contributions.* FB and HF wrote the manuscript. AN designed the experiments. FB carried out OH reactivity measurements and FB and RB carried out TOC measurements. FH, AH and AW contributed to the application of the instrument in the experiments. All co-authors discussed the content of the paper and contributed to the writing.

*Competing interests.* At least one of the (co-)authors is a member of the editorial board of Atmospheric Chemistry and Physics. The authors
declare to have no other competing interests.

*Acknowledgements.*



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
