# Peer review of "Temperature-dependent rate coefficients for the reaction of OH radicals with selected alkanes, aromatic compounds and monoterpenes"

_EGUsphere, 2024_

## Author Response (AR1)

**Responses to the comments of referee #1**

We thank the reviewer for the positive comments.

**Comment:** Line 15 is: „...inorganic and organic in the...” should be "inorganic and organic compounds in the...."

**Response:** We added "compounds" in the text.

**Comment:** Fig. 1 MFC the acronym should be explained.

**Response:** We added "mass flow controller" in the caption.

**Comment:** Line 59 is: "282 nm" should be 266 nm

**Response:** We changed the text accordingly.

**Comment:** Line 71 is: Fig(??) should be Fig 1.

**Response:** We changed the text accordingly.

**Comment:** Line 74 is: 20 Lmin−1), remove the right bracket

**Response:** We changed the text accordingly.

**Comment:** Line 121: was the $k_0$ measured every time before the k measurement? Was it back-to-back? Please explain.

**Response:** The zero decay was measured during the time when the temperature was changed in the flow tube. This is explained in Line 90. We added in Line 120: "The values were measured each time when the temperature of the flow tube was changed."

**Comment:** Line 196 it should go into the $k_0$, anyway

**Response:** This is correct. We added in Line 196: "The small reactivity from this reaction is also included in the value of the zero decay."

**Comment:** Fig 5c. Is it not Dillon et al?

**Response:** We changed the legend of the figure accordingly.

**Comment:** Table 6 and Table 7 is: "GC-FID" GC-FID is an analytical technique, not a kinetic method. It should be the relative rate method (RR). GC-FID should be moved to the figure caption.

**Response:** We changed the columns in Table 5, 6, 7 accordingly.

**Responses to the comments of referee #2**

We thank the reviewer for the positive comments.

**Comment:** Title: Should refer to 'reactions of OH' rather than 'reaction of OH'.

**Response:** We changed the title accordingly.

**Comment:** Line 11: I'm not sure that a fair comparison can be made between uncertainties obtained in a single study and those recommended by IUPAC/NASA-JPL evaluation panels which consider a wide range of studies, and as a result a wider range of values and uncertainties.

**Response:** We agree that this statement may not be fair and cancelled it.

**Comment:** Line 15: 'inorganic and organic in the gas phase...' should be 'inorganic and organic compounds in the gas phase' or similar.

**Response:** We changed the text accordingly.

**Comment:** Line 31: 'only few studies' should be 'only a few studies' or 'there are few studies'.

**Response:** We changed the text accordingly.

**Comment:** Line 48: Is it possible to give approximate values for 'low pressure'.

**Response:** We added "..., e.g. in the range of 100 hPa,..."

**Comment:** Line 58: 282 nm to 266 nm.

**Response:** We changed the text accordingly.

**Comment:** Line 63: The approach to sample from a reaction cell at ambient pressure (or above) into a low pressure detection cell in laboratory experiments has been described in previous work and can lead to sampling issues which impact results. The description of the instrument by Lou et al. reported a negative bias in kinetics at pseudo-first-order

rate coefficients above 60 s-1, which is potentially a result of sampling issues. How does this impact experiments/results described in this work?

**Response:** Such an effect would be visible in a deviation of the shape of the decay curve from a single exponential behaviour at the beginning of the decay (approximately in the first 100 ms) as this effect is associated with an inhomogeneous distribution of the initial OH concentration in the flow tube. In this work, such an effect was not observed in the shape of the decay curve and concentrations of reactants were chosen so that the maximum total reactivity in each series of measurements was less than 20 s$^{-1}$ as described in L152.

**Comment:** Line 71: Reference to Figure 1.

**Response:** We changed the text accordingly.

**Comment:** Line 76: Is the flow condition expected (or required) to be laminar?

**Response:** The flow is laminar as indicated by a small Reynolds number. The measurement method does not necessitate a laminar flow, which, however, helps reducing the OH wall loss and may reduce noise in the measurement. We added in Line 76 that the flow is laminar.

**Comment:** Line 85: Sensors at the inlet and outlet of the tube do not strictly show that there are no temperature gradients along the length of the tube. Have any measurements been made at other points along the tube?

**Response:** The design of the flow tube (comprising a welded double-wall stainless steel tube) does not allow to install sensors within the tube. However, as the temperature was regulated by the temperature-controlled water surrounding the flow tube, it can be reasonably assumed that the temperature within the tube is not significantly different from the sensor readings, provided that the temperatures recorded in the air at the temperature-controlled inlet and outlet are identical.

**Comment:** Figure 2: What causes the observed rise in signal at early times? The production of OH ought to be rapid, does this result from the sampling time from the reaction cell into the detection cell? Is there any impact on the observed kinetics? The fits to the signal shown at longer times do appear to deviate from the observations, particularly at the higher temperatures. It would be helpful to display the decays on a linear y-scale extending to zero to show the extent of the offset, and to provide the fit

results for the decays shown. Further examples of observed decays would also be helpful.

**Response:** The initial few milliseconds are affected by inhomogeneities in the initial OH concentration, which then become more homogeneous as a result of diffusion. As mentioned by the reviewer above, this is commonly observed in this measurement method. However, this does not affect the single exponential behaviour, as this portion of the data is not utilized in the fitting process and thus does not influence the observed kinetics. The discrepancies between the fitted and observed counts at longer times are a consequence of the low count rate, which is insufficient to produce a signal above the noise level. This section solely establishes the offset of the single exponential fit, without influencing the fitted decay time. There is no temperature effect. The discrepancies between the curves are a consequence of the lower absolute counts for the higher measurements at higher temperatures in this example, in which we show normalised counts. It is our contention that the addition of further curves would not yield any new insights for the reader. We believe that the log-scale of the y-axis provides a much better impression of the good exponential behaviour exhibited by the decay curve, which can be observed over 2 orders of magnitude. We added in the caption: "Data from the first few milliseconds are not included in the fit and are affected by inhomogeneities in the initial OH radical concentrations. At low counts, the noise of the detector becomes visible."

**Comment:** Line 114: Please give a typical value for the offset. The data shown in Figure only extend to 0.5 s but the description refers to observations being made for 1 s. In line 148 there is a reference to the decay being observable for 0.4 s. Please clarify.

**Response:** The offset depends on the number of photolysis laser shots that are summed prior to the application of the fit. We added in Line 117 "For these conditions, the typical offset values of the fit were around 12 counts.". In Figure 2, the x-axis range is limited to 0.5s to optimise the range of the plot, as the majority of the normalised counts fell below the low limit of the y-axis of 0.05 in the x-axis range between 0.5s and 1s. The data acquisition, however, lasts for 1s. As illustrated in Figure 2, nearly all OH radicals reacted away within the first 400 ms and only detector noise is measured at later times. As also mentioned in Line 148, this is only the case for the specific conditions of the experiment (reactant concentration, ozone concentration, water vapour concentration, laser power, ...). We clarified this in Line 148 by adding "... for 0.4s before only detector noise was observed ...".

**Comment:** Line 119: The method adopted by the authors appears to involve measurement at a single VOC concentration, which gives relatively low pseudo-first-order rate coefficients on the order of 10-20 s-1, and separate measurement of k0. The

procedure typically adopted in the literature involves measuring pseudo-first-order kinetics for a wide range of VOC concentrations, with pseudo-first-order rate coefficients often varying over several orders of magnitude, and then determining the bimolecular rate coefficient from the slope of the plot of the pseudo-first-order rate coefficients against VOC concentration. The intercept of such plots then represents losses from other processes (i.e. k0), and separate measurement of k0 is not necessary. Please clarify and provide some further rationale for the approach. It would be helpful to discuss the potential uncertainties that could be introduced by measuring a small range of low VOC concentrations and by measuring k0 separately.

**Response:** It is true that additional measurements with varying the VOC concentrations would have avoided using the separate zero decay measurement. However, one of the advantages of the measurement with the instrument used in this work is the low value of the zero decay around $2.5 s^{-1}$, which can be determined with a high precision of approximately $0.2 s^{-1}$ so that its subtraction from the measured rate coefficient in the range of 10 to $20 s^{-1}$ does not add much uncertainty (see also Line 132). In addition, the actual zero decay value was measured immediately before each measurement and showed a high reproducibility. We added in Line 132: "... than the zero-decay rate coefficient, which was around $2.5 s^{-1}$ and was measured with a high precision of $0.2 s^{-1}$, should be used ...".

**Comment:** Line 124: Were any checks for pressure-dependence performed? Were any experiments performed in which experimental conditions such as [O2], laser power, repetition frequency etc. were varied to ensure that the results are not impacted by any experimental conditions?

**Response:** Some experimental conditions, such as an inhomogeneous beam profile of the photolysis laser or flow conditions, can affect the zero decay rate. Therefore, the zero decay rate was measured immediately prior to the measurement of each rate coefficient when conditions were exactly the same. Subtracting this zero decay rate eliminates the possibility that systematic errors due to changing experimental conditions are introduced into the calculation. We tested some of the parameters in the context of another work, where we used a similar instrument for OH reactivity measurements on an aircraft (Fuchs et al, EGUSphere, https://egusphere.copernicus.org/preprints/2024/egusphere-2024-2752/), but none of the parameters tested in that work were relevant to the laboratory measurements in this work.

**Comment:** Line 135: The accuracy of k0 also impacts the results.

**Response:** The accuracy of the zero rate coefficient is given by the reproducibility of its measurement. This is implicitly included in the reproducibility of the rate coefficient measurement because each measurement involves an individual measurement of the zero rate coefficient, so that the variability of this value is part of the reproducibility of the rate coefficient. We added in Line 140: "... , which includes the reproducibility of the zero rate coefficient measurement, ..."

**Comment:** Line 153: How does k0 compare to the values of OH reactivity observed?

**Response:** We addressed this comment in the response to another comment above and added in Line 132: "... than the zero-decay rate coefficient, which was around 2.5 s$^{-1}$ and was measured with a high precision of 0.2 s$^{-1}$, should be used ...".

**Comment:** Line 160: The impact of any impurities depends on their reactivity with OH as well as their concentration, an impurity with high reactivity has the potential to significantly impact results even at low concentrations.

**Response:** We agree that a larger effect cannot be excluded. We added in Line 160: "However, larger effects from small impurities with high reactivity cannot be excluded."

**Comment:** Line 168: Did the TOC measurements agree with the expected VOC concentrations determined from the partial pressures of the gas mixtures prepared?

**Response:** The pressure measurements were not accurate enough to determine the partial pressure of the VOC with a high accuracy. Instead, we validated the method with a certified propane mixture as described in Line 178.

**Comment:** Line 193: Is there any potential impact of O(3P)?

**Response:** O($^3$P) can react with ozone or oxygen slightly changing the ozone concentration. However, as further discussed in the paper the exact ozone concentration does not affect the measurement of the rate coefficients in this work. We are not aware of other O($^3$P) reactions which have the potential to impact the results.

**Comment:** Line 208: What is the OH yield from ozonolysis of myrcene?

**Response:** The OH yield is around 0.7 as discussed in Deng, J. Phys. Chem. A, 2018 doi: 10.1029/1999JD900198. We added this value in Line 215.

**Comment:** Line 213: For the concentrations of VOCs used in this study, how do the concentrations of radicals potentially generated from VOC photolysis compare to the initial OH concentrations used in this study? A fraction of 0.1 % VOC photolysis still has the potential to impact results through radical-radical chemistry if 0.1 % of the initial VOC concentration is similar to the initial concentration of OH.

**Response:** We agree that there may be cases, where photolysis may play a role and should be investigated for the specific case. For the species studied in this work, we do have no indication that this was the case. In general, if an OH radical is formed from the VOC photolysis, this would only increase the initial OH concentration and not affect the measurements. If peroxy radicals are formed, this will increase the reactivity. Rate coefficients can be as high as $1 \times 10^{-10}$ $cm^3s^{-1}$. The resulting reactivity is likely to be in the order of $0.1s^{-1}$, but higher values cannot be excluded. We are not aware of any other reactions that would affect the measurements. We added in Line 213: "If peroxy radicals are formed by photolysis, they may slightly increase the OH loss. For the species studied in this work, there is no indication that photolysis reactions played a role."

**Comment:** Line 215 onwards: Uncertainties should be given for both A and Ea throughout. It would be helpful to tabulate rate coefficients determined for each reaction at each temperature (perhaps in supplementary information).

**Response:** The fitting procedure provides an error value for the A value, which we added in the revised version. However, it is worth noting that the error in the A parameter is highly correlated with the error value of the Ea parameter, so that the total error of the rate coefficient cannot easily be derived from the error propagation of the errors of the 2 fit parameters. Therefore, we only show confidence intervals in the plots instead of an uncertainty band of the fit. The rate coefficients are already listed in the Tables provided in the Supplementary Material, which was submitted together with the manuscript. This is also stated in the data availability statement in the manuscript.

**Comment:** Lines 247 and 250: The use of 'perfect' should be avoided. Where possible it would be good to quantify the agreement with previous results.

**Response:** We changed the text accordingly.

**Comment:** Line 284: 'only few temperatures' to 'only a few temperatures' or 'few temperatures'.

**Response:** We changed the text accordingly.

**Comment:** Figure 4 & line 287: It might be helpful to include the data points from previous work in Figure 4 to illustrate the discussion.

**Response:** We added the data points in Figure 4 and reference the figure again in Line 284 to illustrate the discussion.